

# Federated learning-driven collaborative recommendation system for multi-modal art analysis and enhanced recommendations

Bei Gong[1,2], Ida Puteri Mahsan[2] and Junhua Xiao[1,3]

[1] Department of Art & Design, Gongqing College of Nanchang University, Jiangxi, China
[2] Department of Art & Design, Faculty of Art, Sustainability & Creative Industry, Universiti Pendidikan Sultan Idris, Tanjong Malim, Perak, Malaysia
[3] College of Creative Arts, Universiti Teknologi MARA (UiTM), Shah Alam, Malaysia

## ABSTRACT

With the rapid development of artificial intelligence technology, recommendation systems have been widely applied in various fields. However, in the art field, art similarity search and recommendation systems face unique challenges, namely data privacy and copyright protection issues. To address these problems, this article proposes a cross-institutional artwork similarity search and recommendation system (AI-based Collaborative Recommendation System (AICRS) framework) that combines multimodal data fusion and federated learning. This system uses pre-trained convolutional neural networks (CNN) and Bidirectional Encoder Representation from Transformers (BERT) models to extract features from image and text data. It then uses a federated learning framework to train models locally at each participating institution and aggregate parameters to optimize the global model. Experimental results show that the AICRS framework achieves a final accuracy of 92.02% on the SemArt dataset, compared to 81.52% and 83.44% for traditional CNN and Long Short-Term Memory (LSTM) models, respectively. The final loss value of the AICRS framework is 0.1284, which is better than the 0.248 and 0.188 of CNN and LSTM models. The research results of this article not only provide an effective technical solution but also offer strong support for the recommendation and protection of artworks in practice.

# INTRODUCTION
## Background
With the rapid development of artificial intelligence technology, deep learning-based recommendation systems have been widely applied in various fields. From e-commerce product recommendations to music and video content recommendations, recommendation systems have become important tools to enhance user experience. In the art field, especially in the similarity search and recommendation systems for artworks, there are unique challenges (*Wu et al., 2024*; *Mintie, 2023*). Artworks often have high

Corresponding author
Ida Puteri Mahsan,
idaputeri@fskik.upsi.edu.my

originality and commercial value, making data privacy and copyright protection issues particularly important (*Nishioka, Hauke & Scherp, 2020*; *Xiong & Zhang, 2023*; *Ajmal et al., 2023*).

Freelance artists and art galleries are the main creators and collectors of artworks. They want to increase the exposure and sales of their works through advanced recommendation systems. However, these institutions are reluctant to share their original artwork data directly due to concerns about data breaches and copyright infringement (*White & Matulionyte, 2020*; *Peplow, 2021*). Additionally, wealthy art collectors and museums are also primary consumers of artworks. These entities play a significant role in the art market, seeking advanced recommendation systems to enhance their collections while ensuring privacy and copyright protection.

The extensive use of AI painting tools has also raised concerns about copyright infringement of unpublished works. In recent years, data privacy and copyright protection issues have drawn widespread attention in the art field. Getty Images accused the AI company Stability AI of using millions of unauthorized image data to train its image generation model, which harmed the commercial value of its images (*BakerHostetler, 2024*). In the field of AI-generated artworks, a platform used more than 100,000 artworks without authorization to train its generation model. These works came from thousands of artists, harming the interests of original artists (*Appel, Neelbauer & Schweidel, 2023*). These cases highlight the severe challenges of data privacy and copyright protection in the art field. These issues make it urgent to achieve efficient artwork recommendation while protecting data privacy and copyrights.

This work aims to address these challenges by proposing a federated learning framework that enables multiple institutions to collaboratively train recommendation models without sharing their raw data. By leveraging federated learning, we can enhance the quality of similarity search and recommendation for artworks while ensuring that the data privacy and copyright protection concerns of freelance artists and art galleries are adequately addressed.

## Related work

Numerous studies have explored the application of deep learning and federated learning in recommendation systems, emphasizing the importance of data privacy and copyright protection. For instance, *Chen et al. (2023)* reviewed the latest developments in deep reinforcement learning for recommendation systems but noted insufficient consideration of data privacy issues. *Dong et al. (2022)* investigated trust-aware recommendation systems, focusing on robustness and interpretability, yet their exploration of federated learning was limited. *Lee & Kim (2022)* proposed deep learning recommendation systems using cross-convolution filters to capture complex user-item interactions, though they did not adequately address cross-institution collaboration.

Recent advancements have highlighted the potential of federated learning to enhance recommendation systems while maintaining data privacy. *Dong et al. (2023)* introduced the FedSR framework for Point-of-Interest recommendation systems, addressing data sparsity and Non-Independent and Identically Distributed (Non-IID) issues through

Contrastive Learning. *van Berlo, Saeed & Ozcelebi (2020)* developed a federated unsupervised representation learning architecture, demonstrating how federated learning can pre-train deep neural networks with unlabeled data to protect user privacy. *Elayan, Aloqaily & Guizani (2021)* proposed a Deep Federated Learning framework for decentralized healthcare systems, illustrating the benefits of federated learning in maintaining user privacy and improving model performance in sensitive data environments.To address these issues, more and more researchers are focusing on these problems and seeking innovative solutions. However, previous studies have many shortcomings, as shown in Table 1.

## Our contribution

This study proposes a cross-institutional artwork similarity search and recommendation system—AICRS (AI-based Collaborative Recommendation System) framework, which combines multimodal data fusion and federated learning to address data privacy and copyright protection issues, as shown in Fig. 1. The main contributions of this article are as follows:

Figure 1 illustrates the AICRS framework, consisting of multiple components working together to provide a secure and efficient recommendation system. The framework involves various entities such as art galleries, artists, trading platforms, and self-employed individuals who contribute their data while maintaining control over their local models. The data, which remains with the participants, includes diverse artwork types (advertising, comics, publicity, art collections, *etc.*), represented by different colors for different types. The local models are trained on participant-owned data, and only the necessary parameters are exchanged between participants and the central trusted server. This server aggregates the parameters without exposing the original artwork, ensuring privacy and copyright protection while making customer recommendations based on their needs. The integration of these components ensures a collaborative yet secure environment for recommending artwork across institutions, leveraging the strengths of federated learning and multimodal data fusion.

- We propose a Local Multi-Modal Feature Extraction and Aggregation algorithm (L-MFEA). It combines pre-trained convolutional neural networks (CNN) and Bidirectional Encoder Representations from Transformers (BERT) models. This extracts rich features from image and text data, generating multi-modal feature vectors and improving the accuracy of the recommendation system.
- We propose a Federated Model Parameter Aggregation and Optimization algorithm (F-MPAO). This algorithm trains models locally at each participating institution and aggregates parameters to optimize the global model. It effectively addresses privacy and security issues caused by centralized data storage.
- Combining multimodal data fusion and federated learning strategies, we propose a cross-institutional artwork similarity search and recommendation system (AICRS framework). This improves the overall performance and recommendation effect of the recommendation system while ensuring data privacy and copyright protection.

**Table 1  Summary of related studies.**

| Author | Application scenario | Research content | Possible shortcomings |
|---|---|---|---|
| *Chen et al. (2023)* | Recommendation systems | Review of the application and latest developments of deep reinforcement learning in recommendation systems | Insufficient consideration of data privacy and copyright protection issues |
| *Dong et al. (2022)* | Trust-aware recommendation systems | Trust-aware recommendation systems from the perspective of deep learning, including social trust, robustness, and interpretability | Limited exploration of applications in multimodal data fusion and federated learning |
| *Lee & Kim (2022)* | Recommendation systems | Deep learning recommendation systems based on cross-convolution filters, capturing complex interactions between user and item features | Insufficient protection of data privacy and cross-institution collaboration |
| *Park & Lee (2023)* | Deep sleep recommendation systems | Personalized deep sleep recommendation using hybrid deep learning methods, combining user and collaborative filtering methods | No consideration of data protection and copyright issues |
| *Jeong & Kim (2022)* | Context-aware recommendation systems | Deep learning recommendation systems based on context features, combining neural networks and autoencoders for feature extraction and score prediction | Few applications for data privacy protection and cross-institution data collaboration |
| *Vu & Le (2023)* | Multi-criteria recommendation systems | Context-aware multi-criteria recommendation systems based on deep learning, using deep neural networks to predict ratings and learn aggregation functions | Insufficient consideration of privacy protection and copyright protection |
| *Tegene et al. (2023)* | Collaborative recommendation systems | Latent factor models based on deep learning and embedding to solve data sparsity issues and extract nonlinear features | Limited research on applications in federated learning and multimodal data fusion |
| *Wu, Sun & Shang (2023)* | Deep learning recommendation systems | Proposed DE-Opt framework to optimize hyperparameters of deep learning recommendation systems, improving recommendation accuracy and computational efficiency | Lack of exploration of multimodal data fusion and cross-institution data protection |
| *Torkashvand, Jameii & Reza (2023)* | Collaborative filtering recommendation systems | Systematic review of deep learning collaborative filtering recommendation systems, categorizing and analyzing existing methods and their advantages and disadvantages | Insufficient research on data privacy and cross-institution collaboration protection |
| *Arthur et al. (2022)* | Cross-domain recommendation systems | Proposed a discriminative geometric deep learning model to solve cold start and data sparsity issues in cross-domain recommendations | Less exploration of applications in multimodal data fusion and privacy protection |
| *Dong et al. (2023)* | POI recommendation systems | Proposed FedSR framework for POI-RS using sequential information and Contrastive Learning to address data sparsity and Non-IID issues in FL | Geographic and cultural differences affecting model training effectiveness |
| *van Berlo, Saeed & Ozcelebi (2020)* | Federated learning | Introduced federated unsupervised representation learning for pre-training deep neural networks with unlabeled data in a federated setting | Limited to scenarios where labeled data can be generated from user interaction |
| *Elayan, Aloqaily & Guizani (2021)* | Healthcare systems | Proposed Deep Federated Learning framework for decentralized healthcare systems to maintain user privacy and improve model performance | Model conversion time affecting quality of service to users |

The remainder of this article is organized as follows: "Research Issues" discusses the key research issues related to recommendation systems and federated learning, and model aggregation strategies. "Experiments and Results" presents the experimental setup and results, including a detailed analysis of the system's performance. Finally, "Conclusion" concludes the article and highlights potential directions for future research.
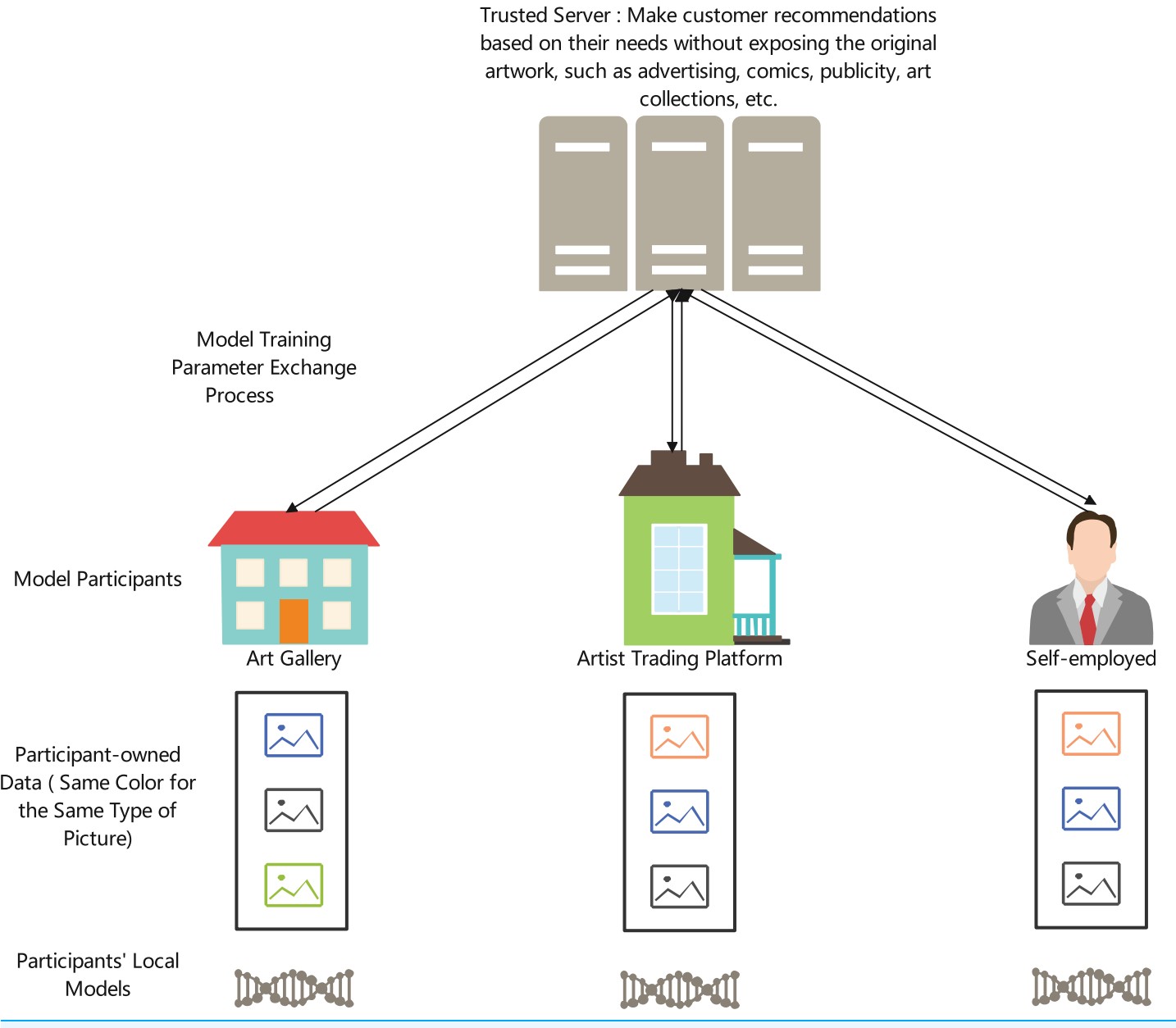

Trusted Server : Make customer recommendations based on their needs without exposing the original artwork, such as advertising, comics, publicity, art collections, etc.

Model Training Parameter Exchange Process

Model Participants

Art Gallery

Artist Trading Platform

Self-employed

Participant-owned Data ( Same Color for the Same Type of Picture)

Participants' Local Models

**Figure 1** **AICRS framework (Image source: Author's own).**

## RESEARCH ISSUES

Protecting user data privacy and copyright is crucial in recommendation systems. Federated learning allows collaborative model training without sharing raw data, ensuring privacy and security. In art recommendation systems, this protects sensitive artwork data and copyrights. Combining federated learning with deep learning, we propose the AICRS (AI-based Collaborative Recommendation System) for efficient and secure artwork recommendations.

## Privacy protection in art recommendation systems

The dataset of artworks used in this research is represented as $\mathscr{D} = \{\mathscr{D}_1, \mathscr{D}_2, \ldots, \mathscr{D}_N\}$. Each $\mathscr{D}_i$ represents a local dataset of a participant (freelance artist or art gallery) containing $n_i$ artworks, $i.e.$, $\mathscr{D}_i = \{(x_{i,j}, y_{i,j})\}_{j=1}^{n_i}$. Here, $x_{i,j}$ represents the feature of the $j$-th artwork of the $i$-th participant, and $y_{i,j}$ represents the corresponding label. Each participant trains their model locally and updates parameters by minimizing the following loss function:

$$\theta_i^{(t+1)} = \theta_i^{(t)} - \eta \nabla_\theta \mathscr{L}_i\left(\theta_i^{(t)}\right) \tag{1}$$

where $\mathscr{L}_i(\theta) = \frac{1}{n_i} \sum_{j=1}^{n_i} \ell(f(x_{i,j}; \theta), y_{i,j})$ is the loss function of the $i$-th participant, $\ell(\cdot, \cdot)$ is the loss function ($e.g.$, mean squared error or cross-entropy), and $\eta$ is the learning rate. Commonly used in stochastic gradient descent, is discussed in various works including (*Jeon et al., 2023*). The global model is updated by aggregating the local model parameters of all participants:

$$\theta^{(t+1)} = \frac{1}{N} \sum_{i=1}^{N} \theta_i^{(t+1)} \tag{2}$$

In the recommendation system, we use deep learning models to extract features of artworks for similarity search and recommendation. Suppose the deep learning model is a convolutional neural network (CNN), its structure can be represented as:

$$h = \text{CNN}(x; \theta) \tag{3}$$

where $h$ is the extracted high-dimensional feature vector, $x$ is the input artwork, and $\theta$ is the model parameter. For similarity search, we use cosine similarity to measure the similarity between two artwork feature vectors:

$$\text{sim}(h_i, h_j) = \frac{h_i \cdot h_j}{||h_i|| ||h_j||} \tag{4}$$

where $h_i$ and $h_j$ represent the feature vectors of the $i$-th and $j$-th artworks, respectively, and $|| \cdot ||$ denotes the $L_2$ norm of the vector. The cosine similarity formula is a widely used method for comparing feature vectors in recommendation systems, as detailed in (*Sankararaman et al., 2020*). Our optimization objective is to maximize the performance of the recommendation system while protecting data privacy and copyrights. Specifically, the optimization objective can be expressed as:

$$\min_\theta \frac{1}{N} \sum_{i=1}^{N} \mathscr{L}_i(\theta) + \lambda \mathscr{R}(\theta) \tag{5}$$

where $\mathscr{R}(\theta)$ is the regularization term, and $\lambda$ is the regularization coefficient. This formula is a standard approach to incorporating regularization in machine learning, discussed extensively in *Tian, Zhang & Zhang (2023)*.

**Problem 1** *Our core research question is how to achieve an efficient artwork recommendation system using federated learning and deep learning technologies while*

protecting the privacy and copyright of artwork data. The specific mathematical model is as follows:

$$\min_{\theta} \frac{1}{N} \sum_{i=1}^{N} \mathcal{L}_i(\theta) + \lambda \mathcal{R}(\theta)$$

$$subject\ to \quad \theta_i^{(t+1)} = \theta_i^{(t)} - \eta \nabla_{\theta} \mathcal{L}_i(\theta_i^{(t)})$$

$$\theta^{(t+1)} = \frac{1}{N} \sum_{i=1}^{N} \theta_i^{(t+1)} \qquad (6)$$

$$h = \mathrm{CNN}(x; \theta)$$

$$sim(h_i, h_j) = \frac{h_i \cdot h_j}{||h_i|| ||h_j||}$$

## Local feature extraction and aggregation based on multimodal data fusion

### Multimodal data fusion: local feature extraction and aggregation based on CNN and BERT models

- Traditional unimodal algorithms, like CNN or LSTM, can only handle single-modal data (image or text). They cannot fully utilize the multimodal features of artworks (*Daneshvar & Ravanmehr, 2022*). They also cannot effectively capture details in artworks, leading to large prediction errors and high loss values, resulting in poor recommendation effects (*Fachrela et al., 2023*). Traditional methods need to store and process large amounts of data centrally, posing data privacy and copyright protection issues (*Nithya, Geetha & Kumar, 2024*).

- The AICRS framework combines multimodal data fusion techniques (image and text features). It uses pre-trained CNN and BERT models to extract rich features from image and text data and then fuses them to generate multimodal feature vectors. This can better capture the details of artworks, significantly reduce prediction errors and loss values, and improve the accuracy of the recommendation system. By using federated learning strategies, model parameters are shared among multiple participating institutions instead of directly sharing data. This protects data privacy and copyright while improving the generalization ability of the model.

### Multimodal data fusion

Suppose there are $N$ artworks. Each artwork $x_i$ contains image data $x_i^{\mathrm{img}}$ and text data $x_i^{\mathrm{text}}$. We first extract features from the image and text separately. For image feature extraction, we use a pre-trained CNN, such as Residual Neural Network (ResNet), to extract the image feature vector:

$$h_i^{\mathrm{img}} = \sigma \left( \sum_{k=1}^{K} \sum_{u=1}^{U} w_k^{(u)} * x_i^{\mathrm{img}} + b_k^{(u)} \right) \qquad (7)$$

where $h_i^{\text{img}}$ represents the image feature vector of the $i$-th artwork, $w_k^{(u)}$ and $b_k^{(u)}$ are the weights and biases of the $k$-th convolution kernel in the $u$-th layer, $*$ denotes the convolution operation, and $\sigma$ denotes the activation function (such as Rectified Linear Unit (ReLU)). Specifically, suppose the CNN contains $L$ layers of convolution and pooling layers. The output for the $l$-th convolution and pooling layer can be represented as:

$$h_i^{(l)} = \sigma\left(\text{Pool}\left(\sum_{m=1}^{M}\sum_{v=1}^{V} w_m^{(v,l)} * h_i^{(l-1)} + b_m^{(v,l)}\right)\right) \tag{8}$$

where $h_i^{(l-1)}$ is the output of the $l-1$ layer, Pool denotes the pooling operation, $w_m^{(v,l)}$ and $b_m^{(v,l)}$ are the weights and biases of the $m$-th convolution kernel in the $v$-th layer. The final image feature vector $h_i^{\text{img}}$ is represented by the output of the last layer of the CNN:

$$h_i^{\text{img}} = \sigma\left(\sum_{n=1}^{N}\sum_{z=1}^{Z} w_n^{(z,L)} * h_i^{(L-1)} + b_n^{(z,L)}\right) \tag{9}$$

For text feature extraction, we use a pre-trained BERT model to extract the text feature vector:

$$h_i^{\text{text}} = \text{BERT}(x_i^{\text{text}}; \theta_{\text{text}}) = \sum_{j=1}^{J}\sum_{q=1}^{Q} \alpha_j^{(q)} h_j^{\text{embed}} + b_j^{(q)} \tag{10}$$

where $h_i^{\text{text}}$ represents the text feature vector of the $i$-th artwork, $\alpha_j^{(q)}$ and $b_j^{(q)}$ are the attention weights and biases of the $j$-th word in the $q$-th layer, and $h_j^{\text{embed}}$ represents the embedding vector of the $j$-th word. Specifically, BERT uses a multi-layer Transformer structure to extract text features. The Transformer architecture in BERT consists of multiple layers of self-attention and feed-forward neural networks, which enables it to capture complex dependencies and contextual information from the text data. For the $l$-th layer of the Transformer, the output can be represented as:

$$h_i^{\text{text}(l)} = \text{LayerNorm}\left(h_i^{\text{text}(l-1)} + \text{MultiHeadAttention}(h_i^{\text{text}(l-1)}; \theta_{\text{attn}}^{(l)})\right) \tag{11}$$

where $h_i^{\text{text}(l-1)}$ is the output of the $l-1$ layer, MultiHeadAttention denotes the multi-head attention mechanism, and $\theta_{\text{attn}}^{(l)}$ are the parameters of the $l$-th layer. The final text feature vector $h_i^{\text{text}}$ is represented by the output of the last layer of BERT:

$$h_i^{\text{text}} = \text{LayerNorm}\left(h_i^{\text{text}(L-1)} + \text{MultiHeadAttention}(h_i^{\text{text}(L-1)}; \theta_{\text{attn}}^{(L)})\right) \tag{12}$$

To generate a comprehensive multimodal feature vector, we concatenate the image and text features extracted from CNN and BERT, then process them through a fully connected layer. The feature fusion process can be represented as:

$$h_i = \sigma\left(W_{\text{fusion}}\begin{bmatrix} h_i^{\text{img}} \\ h_i^{\text{text}} \end{bmatrix} + b_{\text{fusion}}\right) \tag{13}$$

**Peer**J Computer Science

where $h_i$ represents the multimodal feature vector of the $i$-th artwork, $[h_i^{\text{img}}, h_i^{\text{text}}]$ represents the concatenation of image and text features, $W_{\text{fusion}}$ is the weight matrix, $b_{\text{fusion}}$ is the bias vector, and $\sigma$ represents the activation function. Suppose the fully connected layer contains $M$ neurons. The computation process can be represented as:

$$h_i^{(f)} = \sigma\left(\sum_{k=1}^{K} \sum_{r=1}^{R} W_k^{(r)} \begin{bmatrix} h_{i,k}^{\text{img}} \\ h_{i,k}^{\text{text}} \end{bmatrix} + b_k^{(r)}\right) \tag{14}$$

where $h_i^{(f)}$ represents the output of the fully connected layer, $W_k^{(r)}$ and $b_k^{(r)}$ are the weights and biases of the $k$-th neuron in the $r$-th layer, and $\sigma$ represents the activation function (such as ReLU). The final multimodal feature vector $h_i$ can be represented as:

$$h_i = \sigma\left(\sum_{k=1}^{K} \sum_{r=1}^{R} W_k^{(r)} \begin{bmatrix} h_{i,k}^{\text{img}} \\ h_{i,k}^{\text{text}} \end{bmatrix} + b_k^{(r)}\right) \tag{15}$$

We further normalize the feature vector $h_i$ to ensure its balance between different modalities:

$$\hat{h}_i = \frac{h_i}{||h_i||_2} = \frac{h_i}{\sqrt{\sum_{j=1}^{J} h_{i,j}^2 + \varepsilon}} \tag{16}$$

where $||h_i||_2$ represents the $L_2$ norm of $h_i$, and $\varepsilon$ is a small constant to avoid division by zero. This normalization ensures that the length of each feature vector is on a uniform scale, improving the accuracy of subsequent processing and recommendation. The output $\{\hat{h}_i\}_{i=1}^{N}$ of this algorithm will be used as input for the subsequent recommendation system model.

## Federated learning-based cross-institutional model parameter aggregation and optimization

### Cross-institutional collaborative model: parameter aggregation and optimization based on federated learning framework

- In cross-institutional data sharing, traditional methods cannot effectively protect the data ownership and copyright of each participating institution. This makes institutions reluctant to share data, affecting the overall performance of the model. Traditional model training methods require huge computational and storage resources when handling large-scale data. This can lead to single point failures and poor system robustness (*Wang & Kawagoe, 2018*; *Musto et al., 2010*; *Kim, Kang & Lee, 2019*). Traditional algorithms find it difficult to achieve efficient model parameter updates and optimization in cross-institutional cooperation, leading to slow convergence of the global model and limited performance improvement (*Messina et al., 2019*; *Wang et al., 2019*).

- This algorithm aggregates local model parameters of each participating institution through methods such as weighted averaging. It ensures data ownership and copyright of

each institution, enhances their willingness to cooperate, and improves the overall performance of the model. This algorithm reduces reliance on a central server, improves system robustness, and avoids single point failures. With improved optimization algorithms and effective parameter aggregation strategies, it significantly speeds up the convergence of the global model and improves performance.

### Cross-institutional collaborative model

Suppose there are $N$ participating institutions. Each institution $i$ has a local dataset $\mathscr{D}_i = \{(x_{i,j}, y_{i,j})\}_{j=1}^{n_i}$, where $x_{i,j}$ is the $j$-th sample of the $i$-th institution, and $y_{i,j}$ is its corresponding label. On the local server of each participating institution, use the L-MFEA algorithm to extract multimodal features $h_{i,j}$:

$$h_{i,j} = \text{L-MFEA}(x_{i,j}; \theta_i) \tag{17}$$

where $\theta_i$ is the local model parameter of the $i$-th institution. Next, train the model on the local dataset to minimize the local loss function:

$$\mathscr{L}_i(\theta_i) = \frac{1}{n_i} \sum_{j=1}^{n_i} \ell\big(f\big(h_{i,j}; \theta_i\big), y_{i,j}\big) + \lambda||\theta_i||_2^2 \tag{18}$$

where $\ell(\cdot, \cdot)$ is the loss function (such as cross-entropy loss), $f$ is the model output, and $\lambda||\theta_i||_2^2$ is the regularization term.

Local model parameters are updated using the gradient descent algorithm:

$$\theta_i^{(t+1)} = \theta_i^{(t)} - \eta\left(\nabla_{\theta_i}\mathscr{L}_i\big(\theta_i^{(t)}\big) + \gamma\nabla_{\theta_i}\mathscr{R}\big(\theta_i^{(t)}\big)\right) \tag{19}$$

where $\eta$ is the learning rate, $\mathscr{R}(\theta_i)$ is the regularization term, and $\gamma$ is the weight of the regularization term. In each round of federated learning, the central server aggregates the local model parameters of each participating institution. The aggregation method, such as weighted averaging, is defined as:

$$\theta^{(t+1)} = \frac{1}{\sum\limits_{i=1}^{N} n_i} \sum_{i=1}^{N} n_i \theta_i^{(t+1)} \tag{20}$$

where $\theta^{(t+1)}$ is the global model parameter, and $n = \sum_{i=1}^{N} n_i$ is the total amount of data from all participating institutions. The central server distributes the aggregated global model parameters $\theta^{(t+1)}$ back to each participating institution to update the local model parameters:

$$\theta_i^{(t+1)} = \theta^{(t+1)} \quad \text{for all} \quad i = 1, 2, \ldots, N \tag{21}$$

Each participating institution, after receiving the updated global model parameters, continues to train on the local dataset to minimize the local loss function. Repeat local training and parameter aggregation until the global model converges.

**Theorem 1** *Given a reasonable learning rate $\eta$ and sufficient iterations, the loss function $\mathscr{L}(\theta)$ of the global model parameter $\theta$ will converge, assuming the local loss functions $\mathscr{L}_i(\theta_i)$ of all participating institutions converge. Specifically, assuming each local model satisfies the following condition during the iteration process:*

$$\mathscr{L}(\theta^{(t+1)}) \leq \mathscr{L}(\theta^{(t)}) - \eta\left(\frac{1}{N}\sum_{i=1}^{N}\left|\left|\nabla_{\theta_i}\mathscr{L}_i\left(\theta_i^{(t)}\right)\right|\right|^2 + \gamma\sum_{i=1}^{N}\left|\left|\nabla_{\theta_i}\mathscr{R}\left(\theta_i^{(t)}\right)\right|\right|^2\right)$$
$$+ \frac{\eta^2}{2}(L_{\mathscr{L}} + L_{\mathscr{R}}) \leq \mathscr{L}^* \tag{22}$$

*where $L_{\mathscr{L}}$ and $L_{\mathscr{R}}$ are the Lipschitz constants of the loss function $\mathscr{L}$ and the regularization term $\mathscr{R}$, respectively, and $\mathscr{L}^*$ is the global optimal loss value.*

**Corollary 1** *Based on the above theorem, if the multimodal feature vector $h_i$ is obtained by concatenating the image feature vector $h_i^{img}$ and the text feature vector $h_i^{text}$ and then computing through the fully connected layer, the final model parameter update can be represented as:*

$$\theta_i^{(t+1)} = \theta_i^{(t)} - \eta\left(\sum_{v=1}^{V}\nabla_{\theta_i}\ell\left(f\left(\sum_{p=1}^{P}\sum_{q=1}^{Q}W_{p,q}^{(img)}h_{i,p}^{img} + W_{p,q}^{(text)}h_{i,q}^{text} + b_{p,q}\right), y_{i,v}\right) + \lambda\theta_i\right) \tag{23}$$

*where $\eta$ represents the learning rate, $W_{p,q}^{(img)}$ and $W_{p,q}^{(text)}$ represent the weight matrices of the image and text features, respectively, $b_{p,q}$ represents the bias vector, and $\lambda$ represents the regularization parameter.*

## AICRS framework: algorithm pseudocode and complexity analysis

The AICRS framework is presented as a comprehensive solution that integrates multiple components and processes to achieve efficient and secure artwork recommendations (Algorithm 1). While the overall structure is defined as a framework, the detailed implementation of its components and processes is expressed in the form of an algorithm. This approach allows us to provide a clear and precise description of the operational steps involved in the framework, ensuring reproducibility and facilitating practical application. By presenting the pseudocode, we aim to illustrate the exact sequence of operations, including data handling, model training, and federated learning procedures, thus bridging the conceptual framework with its practical execution.

Suppose each participating institution has $n_i$ samples. The constant time for image feature extraction and text feature extraction are $C_{img}$ and $C_{text}$, respectively. The time complexity for each training iteration is $C_{train}$, the number of federated learning rounds is $T$, and the number of model parameters is $P$. Considering these factors, the time complexity is $O(T \cdot N^2 \cdot n_i \cdot I)$. For space complexity, each participating institution stores the local dataset $\mathscr{D}_i$ and model parameters $\theta_i$, and the central server stores the global model parameters $\theta$ and the model parameters of each participating institution. Suppose the dimension of each sample is $D$. The space complexity is $O(N \cdot n_i \cdot D + N^2 \cdot P)$.

To further analyze the performance of the proposed framework, we compared the AICRS framework with other state-of-the-art models, as shown in Table 2. The time

---

**Algorithm 1** AICRS framework.

**Input** Local dataset $\mathscr{D}_i = \{(x_{i,j}, y_{i,j})\}_{j=1}^{n_i}$

**Output** Global model parameters $\theta$

**1 for** *each participating institution $i = 1$ to $N$* **do**

**2**        **for** *each sample $j = 1$ to $n_i$* **do**

**3**            Extract multimodal features $h_{i,j}$ using Eq. (17);

**4**            Extract image features $h_i^{\text{img}}$ using Eq. (7);

**5**            Extract text features $h_i^{\text{text}}$ using Eq. (10);

**6**            Concatenate image and text features to generate multimodal feature vector $h_i$ using Eq. (13);

**7**            Normalize the feature vector $h_i$ to obtain $\hat{h}_i$ using Eq. (41);

**8**        Train the model on the local dataset, minimizing the local loss function $\mathscr{L}_i(\theta_i)$ using Eq. (55);

**9**        Update local model parameters $\theta_i^{(t+1)}$ using Eq. (65);

**10 for** *each round of federated learning $t = 1$ to $T$* **do**

**11**        The central server aggregates the local model parameters of each participating institution using Eq. (48);

**12**        Distribute the aggregated global model parameters $\theta^{(t+1)}$ back to each participating institution using Eq. (21);

**13**        **for** *each participating institution $i = 1$ to $N$* **do**

**14**            Continue training on the local dataset, minimizing the local loss function $\mathscr{L}_i(\theta_i)$;

**15**            Update local model parameters $\theta_i^{(t+1)}$;

**16 while** *the global model has not converged* **do**

**17**        **for** each participating institution $i = 1$ to $N$ **do**

**18**            Continue training on the local dataset, minimizing the local loss function $\mathscr{L}_i(\theta_i)$ using Eq. (55);

**19**            Update local model parameters $\theta_i^{(t+1)}$ using Eq. (65);

**20**        The central server aggregates the local model parameters of each participating institution using Eq. (48);

**21**        Distribute the aggregated global model parameters back to each participating institution using Eq. (21);

**22 return** *Global model parameters $\theta$*

---

**Table 2 Comparison of time and space complexities.**

| Related study | Time complexity | Space complexity |
|---|---|---|
| *Karayel (2023)* | $O(\varepsilon^{-2} \ln(\delta^{-1}) + \ln n)$ | $O(\varepsilon^{-2} \ln(\delta^{-1}) + \ln n)$ |
| *Liu & Yu (2022)* | $O(2^{n/\log n})$ | $O(2^n)$ |
| *Malandrino et al. (2021)* | $O(N^3)$ | Variable |
| *Zhou et al. (2023)* | $O(P \log P)$ | $O(N \cdot P)$ |
| AICRS | $O(T \cdot N^2 \cdot n_i \cdot I)$ | $O(N \cdot n_i \cdot D + N^2 \cdot P)$ |

complexity of AICRS is $O(T \cdot N^2 \cdot n_i \cdot I)$, which has better scalability compared to *Karayel*'s *(2023)* $O(\varepsilon^{-2} \ln(\delta^{-1}) + \ln n)$ and *Liu & Yu*'s *(2022)* $O(2^{n/\log n})$. Especially when handling large-scale data and multiple institutions, AICRS is more efficient. Additionally, the space complexity of AICRS is $O(N \cdot n_i \cdot D + N^2 \cdot P)$, which is much lower than *Liu & Yu*'s *(2022)* $O(2^n)$ and *Karayel*'s *(2023)* $O(\varepsilon^{-2} \ln(\delta^{-1}) + \ln n)$. This significantly reduces storage requirements while maintaining performance. Overall, the optimization in time and space complexity of the AICRS algorithm makes it more advantageous in large-scale distributed federated learning systems.

# EXPERIMENTS AND RESULTS

This section describes the dataset and experimental parameters used in our study. It provides an overview of the SemArt dataset, details the data split for training, validation, and testing, and outlines the experimental parameters set for the training and evaluation of the recommendation models. Additionally, it presents the results of the AICRS framework application, highlighting its performance compared to traditional CNN and LSTM models.

## Dataset and experimental parameters

The SemArt dataset is specifically designed for the analysis and recommendation of artworks. It contains 21,384 images of artworks and their related textual descriptions. Each image comes with detailed text descriptions, including the title, artist, creation year, art style, and descriptive text (https://doi.org/10.17036/researchdata.aston.ac.uk.00000380).

To train and test the recommendation system, we split the dataset by art style. Each art style acts as a client for federated learning training. The training set includes a portion of the artworks from each style for model training. The test set includes the remaining artworks for evaluating the model's recommendation accuracy. The accuracy of the recommendations is evaluated by comparing whether the recommended artworks match the actual artist's style and type. The specific data content is shown in Fig. 2.

The dataset is split into training, validation, and test sets with the following percentages: 70% for training, 15% for validation, and 15% for testing. This split ensures that the model is adequately trained, validated, and tested to achieve reliable performance metrics. Our experiments were conducted using high-performance hardware to ensure efficient computation and model training.

Our experimental parameters are set as shown in Table 3. After setting these detailed experimental parameters, we trained and tested these algorithms on the SemArt dataset. The SemArt dataset includes images of artworks and their detailed textual descriptions from various art categories, covering multiple art styles from the 11th to the 20th century. By conducting experiments on this dataset, we can evaluate the effectiveness and performance of the AICRS framework in processing and recommending artworks.

## AICRS framework application results

This study focuses on developing a cross-institutional artwork recommendation system based on federated learning. It uses multimodal data fusion (image and text features) to enhance the performance of the recommendation system. The model accuracy measures

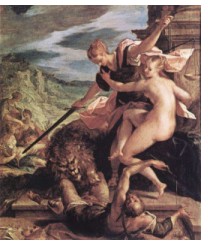

IMAGE_FILE: 00000-allegory.jpg
DESCRIPTION: The painting, displaying the characteristic Mannerist style of the artist, is also known as "The Triumph of Justice".
AUTHOR: Hans von Aachen
TITLE: Allegory
TECHNIQUE: Oil on canvas
DATE: 1598
TYPE: Mythological
SCHOOL: German
TIMEFRAME: 1601-1650

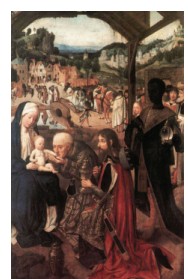

IMAGE_FILE: 15104-magi.jpg
DESCRIPTION: The left side of the painting was cut; originally Saint John was depicted behind Mary; and the motive of hand kissing was in the centre of the composition.
AUTHOR: GEERTGEN tot Sint Jans
TITLE: Adoration of the Magi
TECHNIQUE: Panel
DATE: 1480-85
TYPE: religious
SCHOOL: Netherlandish
TIMEFRAME: 1451-1500

(a)                                                                    (b)

**Figure 2 SemArt dataset samples: The presence of multiple types of art images in the dataset and the corresponding comments and attributes (Image source: SemArt Dataset (https://doi.org/10.17036/researchdata.aston.ac.uk.00000380), licensed under CC BY-NC 4.0.).**

**Table 3 Detailed experimental parameters.**

| Parameter name | Parameter value | Parameter name | Parameter value |
|---|---|---|---|
| Dataset | SemArt | Number of images | 21,384 |
| Image resolution | 224 × 224 | Text description | Each image includes title, artist, creation year, art style, descriptive text |
| Categories | Painting, sculpture, photography, *etc.* | Time span | 11th to 20th Century |
| Art styles | Baroque, renaissance, impressionism, modern art, *etc.* | Model structure (Image Feature Extraction) | ResNet-50 |
| Input size (Image feature extraction) | 224 × 224 × 3 | Number of convolution layers (Image feature extraction) | 50 |
| Activation function (Image feature extraction) | ReLU | Model structure (Text feature extraction) | BERT-base |
| Number of hidden layers (Text feature extraction) | 12 | Number of hidden units (Text feature extraction) | 768 |
| Number of attention heads (Text feature extraction) | 12 | Input length (Text feature extraction) | 128 |
| Number of layers (Feature Fusion) | 2 | Number of neurons (Feature fusion) | 512, 256 |
| Activation function (Feature fusion) | ReLU | Local training epochs | 10 |
| Global training epochs | 50 | Learning rate | 0.001 |
| Optimization algorithm | Adam | Batch size | 32 |

the ability of the recommendation system to recommend the correct artworks, while the loss value reflects the prediction error during training.

Figure 3 shows the accuracy performance of the four models over 50 training rounds. The results indicate that the AICRS framework has significant advantages in processing and recommending artworks, surpassing traditional CNN and LSTM models, as well as the Federated Averaging (FED-AVG) model. The CNN model's final accuracy after 50

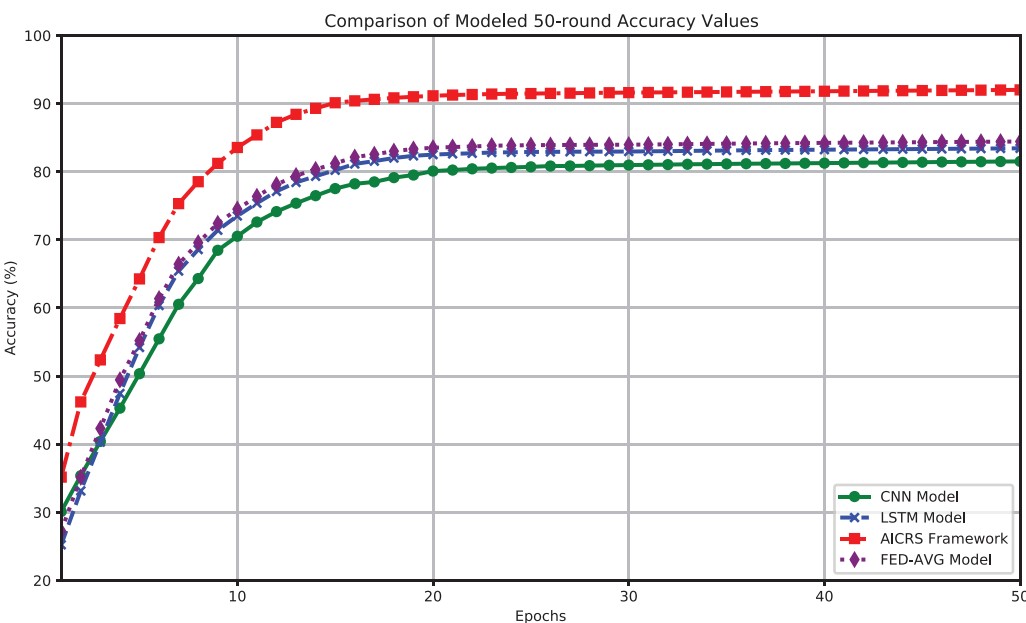

**Figure 3 Comparison of modeled 50-round ACCURACY values.**

training rounds is 81.52%. Accuracy grows rapidly in the early stages but stabilizes, maintaining above 80% from round 25 and reaching 81.52% at the end. The LSTM model's final accuracy after 50 training rounds is 83.44%. The LSTM model also shows rapid accuracy growth initially, reaching about 82.63% around round 22, then slightly improving to 83.44%. The FED-AVG model's final accuracy after 50 training rounds is 84.44%. The FED-AVG model demonstrates strong performance, surpassing the LSTM model in the later stages, with accuracy steadily increasing and reaching 84.44% at the end. The AICRS framework's final accuracy after 50 training rounds is 92.02%. The AICRS framework performs significantly better than CNN, LSTM, and FED-AVG from the start, exceeding 90% accuracy before round 30 and reaching 92.02% at the end.

Figure 4 shows the loss values of the four models (CNN, LSTM, AICRS framework, and FED-AVG) over 50 training rounds. The results show that the AICRS framework has significantly lower loss values in processing and recommending artworks compared to traditional CNN and LSTM models, as well as the FED-AVG model. Specifically, the CNN model's final loss value after 50 training rounds is 0.248. The loss value decreases rapidly in the early stages but stabilizes, staying below 0.26 from round 30 and reaching 0.248 at the end. The LSTM model's final loss value after 50 training rounds is 0.188. The LSTM model also shows a rapid decrease in loss value initially, reaching about 0.22 around round 25, then slightly decreasing to 0.188. The FED-AVG model's final loss value after 50 training rounds is 0.168. The FED-AVG model performs better than the LSTM model, showing a steady decrease in loss value and reaching 0.168 at the end. The AICRS framework's final loss value after 50 training rounds is 0.1284. The AICRS framework performs significantly better than CNN, LSTM, and FED-AVG from the start, exceeding 0.20 loss value before

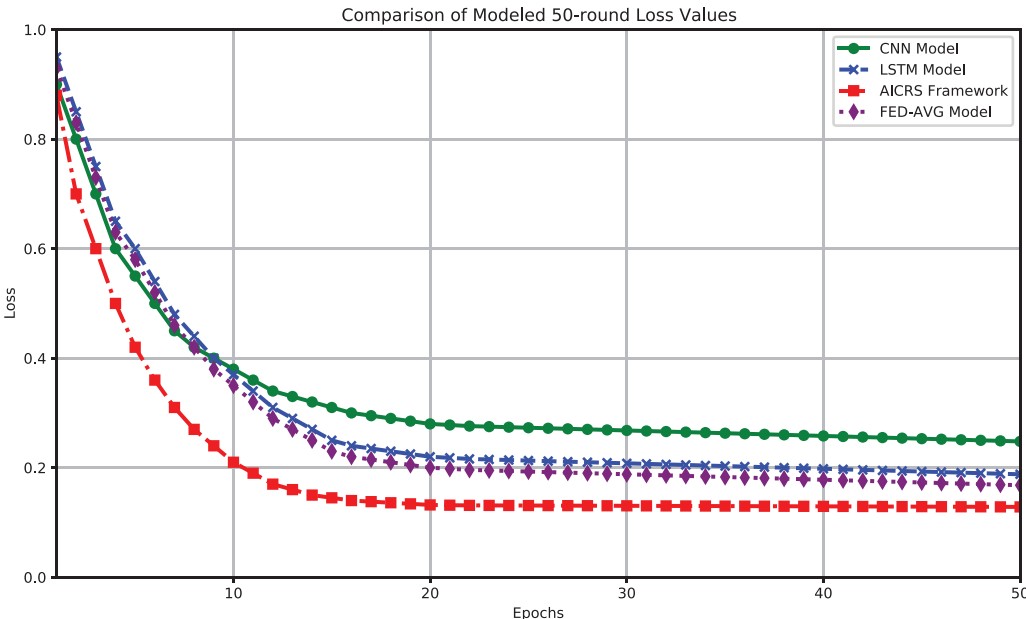

**Figure 4 Comparison of modeled 50-round loss values.**

**Table 4 Model performance comparison.**

| Model | Accuracy (%) | Loss value | Training time (ms) |
|---|---|---|---|
| CNN | 81.52 | 0.248 | 3,962 |
| LSTM | 83.44 | 0.188 | 4,185 |
| AICRS | 92.02 | 0.1284 | 4,577 |

round 20 and reaching 0.1284 at the end. The AICRS framework shows significant advantages in accuracy and loss values, improving recommendation performance and reducing prediction errors in the artwork recommendation system.

Table 4 shows the performance comparison of the three models, including accuracy, loss value and training time. It is evident that the AICRS framework outperforms the CNN and LSTM models in accuracy and loss value. Although its training time is slightly longer, the performance improvement is significant. The training time for each model was measured by recording the duration from the start to the completion of the training process on the same hardware configuration. Specifically, we used a server equipped with dual NVIDIA Tesla V100 GPUs, each with 32 GB of memory, and 256 GB of system RAM. The server is powered by dual Intel Xeon Gold 6258R CPUs, each with 28 cores, providing ample processing power for both training and inference phases. Each model was trained using the same dataset, batch size, and number of epochs to ensure a fair comparison. The reported training times are the average of three runs to account for any variability in the training process. The slight increase in training time for the AICRS framework is justified by its superior performance metrics, indicating a worthwhile trade-off for the significant gains in accuracy and reduced loss value.

## Discussion

The rapid development of artificial intelligence has led to the widespread use of recommendation systems in various fields. In the art sector, implementing similarity search and recommendation systems poses unique challenges due to the high originality and commercial value of artworks. This necessitates robust data privacy and copyright protection. Freelance artists and art galleries aim to increase the exposure and sales of their works through advanced recommendation systems but are often hesitant to share their original artwork data due to concerns about data breaches and copyright infringement.

This study proposes a cross-institutional artwork similarity search and recommendation system (AICRS framework) that combines multimodal data fusion and federated learning to address data privacy and copyright protection issues. The experimental results show that the AICRS framework has significant advantages in processing and recommending artworks, surpassing traditional CNN and LSTM models. The main reasons for the performance differences among the models are as follows:

- The AICRS framework combines multimodal data fusion (image and text features), extracting richer features of artworks compared to single-modal CNN and LSTM models. This improves the accuracy of the recommendation system. The CNN and LSTM models' final accuracy after 50 training rounds are 81.52% and 83.44%, respectively. The AICRS framework's final accuracy reaches 92.02%. This shows that multimodal data fusion has significant advantages in capturing the details and features of artworks.
- The AICRS framework uses pre-trained ResNet-50 and BERT models, which have better feature extraction and representation capabilities than traditional CNN and LSTM models. These models capture the details of artworks more effectively. In terms of loss value, the AICRS framework also performs better than other models. The CNN and LSTM models' final loss values after 50 training rounds are 0.248 and 0.188, respectively. The AICRS framework's final loss value is 0.1284. The more complex model structure enables the AICRS framework to handle complex features and reduce errors more effectively.
- The AICRS framework adopts federated learning strategies, sharing model parameters among multiple participating institutions instead of directly sharing data. This improves the model's generalization ability and privacy protection. This strategy not only protects the data privacy of participating institutions but also enhances the overall performance of the model. The training time of the AICRS framework is slightly longer than other models (4,577 ms), but the performance improvement is significant.

## CONCLUSION

This article proposes a cross-institutional artwork similarity search and recommendation system (AICRS framework) that combines multimodal data fusion and federated learning to address data privacy and copyright protection issues. This system uses pre-trained convolutional neural networks (CNN) and BERT models to extract rich features from

image and text data. It trains models locally at each participating institution and aggregates parameters through a federated learning framework to optimize the global model. The experimental results demonstrate that the AICRS framework has significant advantages in processing and recommending artworks, improving recommendation performance and reducing prediction errors. It enables collaboration among art institutions, offering accurate recommendations and complying with data protection regulations.The AICRS framework still has room for improvement in its reliance on high-quality multimodal data. Future research will explore enhancing the system's robustness in cases of incomplete or low-quality data, as well as expanding the framework to support real-time recommendations across more diverse types of art media.

# APPENDIX: MATHEMATICAL THEOREMS AND COROLLARY PROOFS

**Theorem 2** Let $h_i^{\text{img}}$ and $h_i^{\text{text}}$ be the image and text feature vectors, respectively. $[h_i^{\text{img}}, h_i^{\text{text}}]$ is their concatenation, and $h_i$ is the multimodal feature vector. The normalized multimodal feature vector $\hat{h}_i$ satisfies:

$$\hat{h}_i = \frac{\sigma\left(W_{\text{fusion}}\begin{bmatrix}h_i^{\text{img}}\\h_i^{\text{text}}\end{bmatrix} + b_{\text{fusion}}\right)}{\sqrt{\sum_{k=1}^{K}\left(\sum_{j=1}^{J}\sigma\left(\sum_{m=1}^{M}W_{m,j}^{(\text{img})}h_{i,m}^{\text{img}} + \sum_{n=1}^{N}W_{n,j}^{(\text{text})}h_{i,n}^{\text{text}} + b_{j,\text{fusion}}\right)\right)_k^2 + \varepsilon}} \tag{24}$$

*where $\sigma$ is the activation function, $W_{\text{fusion}}$ is the weight matrix of the fully connected layer, $b_{\text{fusion}}$ is the bias vector, and $\varepsilon$ is a small constant to avoid division by zero.*

**Proof 1** *First, define the optimization problem as follows:*

$$\min_{\{h_i\}_{i=1}^{N}} \sum_{i=1}^{N}||h_i - \hat{h}_i||_2^2 + \lambda\sum_{i=1}^{N}||h_i||_2^2 \tag{25}$$

*where $\lambda > 0$ is the regularization parameter.*

*Expand the objective function as:*

$$J(\{h_i\}_{i=1}^{N}) = \sum_{i=1}^{N}\left(||h_i - \hat{h}_i||_2^2 + \lambda||h_i||_2^2\right) \tag{26}$$

*We can consider each term individually, i.e., optimize for each $h_i$. For each $h_i$, the optimization problem is:*

$$\min_{h_i}||h_i - \hat{h}_i||_2^2 + \lambda||h_i||_2^2 \tag{27}$$

*Expand the above expression and take the derivative:*

$$\frac{\partial}{\partial h_i}\left(||h_i - \hat{h}_i||_2^2 + \lambda||h_i||_2^2\right) = 2(h_i - \hat{h}_i) + 2\lambda h_i \tag{28}$$

*Set the derivative to zero, we get:*

$$2(h_i - \hat{h}_i) + 2\lambda h_i = 0 \Rightarrow h_i(1 + \lambda) = \hat{h}_i \tag{29}$$

*Solve to get:*

$$h_i = \frac{\hat{h}_i}{1 + \lambda} \tag{30}$$

*To obtain the normalized multimodal feature vector $\hat{h}_i$, we normalize the feature vector $h_i$:*

$$\hat{h}_i = \frac{h_i}{||h_i||_2} = \frac{h_i}{\sqrt{\sum_{j=1}^{J} h_{i,j}^2 + \varepsilon}} \tag{31}$$

*where $||h_i||_2$ is the $L_2$ norm of $h_i$, and $\varepsilon$ is a small constant to avoid division by zero.*

*By definition, we have:*

$$h_i = \sigma\left( W_{\text{fusion}} \begin{bmatrix} h_i^{\text{img}} \\ h_i^{\text{text}} \end{bmatrix} + b_{\text{fusion}} \right) \tag{32}$$

*Substitute $\frac{\hat{h}_i}{1+\lambda}$ into the normalization equation and combine with the fusion equation, we get the final normalized multimodal feature vector:*

$$\hat{h}_i = \frac{\sigma\left( W_{\text{fusion}} \begin{bmatrix} h_i^{\text{img}} \\ h_i^{\text{text}} \end{bmatrix} + b_{\text{fusion}} \right)}{\sqrt{\sum_{k=1}^{K} \left( \sum_{j=1}^{J} \sigma\left( \sum_{m=1}^{M} W_{m,j}^{(\text{img})} h_{i,m}^{\text{img}} + \sum_{n=1}^{N} W_{n,j}^{(\text{text})} h_{i,n}^{\text{text}} + b_{j,\text{fusion}} \right) \right)_k^2 + \varepsilon}} \tag{33}$$

*In summary, the optimization problem has a unique solution that satisfies the equation, and the theorem is proved.*

**Corollary 2** *Based on the above theorem, if the multimodal feature vector $h_i$ is obtained by concatenating the image feature vector $h_i^{img}$ and the text feature vector $h_i^{text}$ and then computing through a fully connected layer, the normalized multimodal feature vector $\hat{h}_i$ can be expressed as follows:*

$$\hat{h}_i = \frac{\sigma\left( \sum_{p=1}^{P} \sum_{q=1}^{Q} W_{p,q}^{\text{fusion}} \begin{bmatrix} h_{i,p}^{\text{img}} \\ h_{i,q}^{\text{text}} \end{bmatrix} + b_{p,q}^{\text{fusion}} \right)}{\sqrt{\sum_{r=1}^{R} \left( \sum_{s=1}^{S} \sigma\left( \sum_{t=1}^{T} W_{t,s}^{(\text{img})} h_{i,t}^{\text{img}} + \sum_{u=1}^{U} W_{u,s}^{(\text{text})} h_{i,u}^{\text{text}} + b_{s,\text{fusion}} \right) \right)_r^2 + \varepsilon}} \tag{34}$$

*where $\sigma$ is the activation function, $W_{p,q}^{\text{fusion}}$ is the weight matrix of the fully connected layer, $b_{p,q}^{\text{fusion}}$ is the bias vector, and $\varepsilon$ is a small constant to avoid division by zero.*

**Proof 2** *First, consider the calculation process of the multimodal feature vector $h_i$. Assume $h_i$ is obtained by concatenating the image feature vector $h_i^{\text{img}}$ and the text feature vector $h_i^{\text{text}}$, and then computing through a fully connected layer:*

$$h_i = \sigma\left(W_{\text{fusion}}\begin{bmatrix} h_i^{\text{img}} \\ h_i^{\text{text}} \end{bmatrix} + b_{\text{fusion}}\right) \tag{35}$$

*where $W_{\text{fusion}}$ is the weight matrix of the fully connected layer, $b_{\text{fusion}}$ is the bias vector, and $\sigma$ is the activation function.*

*We further refine the calculation process of the image feature vector $h_i^{\text{img}}$ and the text feature vector $h_i^{\text{text}}$. Assume the image feature vector $h_i^{\text{img}}$ is obtained through a convolutional neural network:*

$$h_i^{\text{img}} = \sigma\left(\sum_{m=1}^{M} W_m^{(\text{img})} x_i^{\text{img}} + b_m^{(\text{img})}\right) \tag{36}$$

*where $W_m^{(\text{img})}$ and $b_m^{(\text{img})}$ are the weights and biases of the convolution kernels, and $x_i^{\text{img}}$ is the input image data.*

*Similarly, assume the text feature vector $h_i^{\text{text}}$ is obtained through a pre-trained BERT model:*

$$h_i^{\text{text}} = \text{BERT}(x_i^{\text{text}}; \theta_{\text{text}}) = \sigma\left(\sum_{n=1}^{N} W_n^{(\text{text})} x_i^{\text{text}} + b_n^{(\text{text})}\right) \tag{37}$$

*where $W_n^{(\text{text})}$ and $b_n^{(\text{text})}$ are the weights and biases of the BERT model, $x_i^{\text{text}}$ is the input text data, and $\theta_{\text{text}}$ is the parameter of the BERT model.*

*Next, we concatenate the image feature vector $h_i^{\text{img}}$ and the text feature vector $h_i^{\text{text}}$ to obtain the multimodal feature vector $h_i$:*

$$h_i = \begin{bmatrix} h_i^{\text{img}} \\ h_i^{\text{text}} \end{bmatrix} \tag{38}$$

*Substitute Eqs. (36) and (37) into (38), we get:*

$$h_i = \begin{bmatrix} \sigma\left(\sum_{m=1}^{M} W_m^{(\text{img})} x_i^{\text{img}} + b_m^{(\text{img})}\right) \\ \sigma\left(\sum_{n=1}^{N} W_n^{(\text{text})} x_i^{\text{text}} + b_n^{(\text{text})}\right) \end{bmatrix} \tag{39}$$

*Then, compute through the fully connected layer to obtain the fused multimodal feature vector $h_i$:*

$$h_i = \sigma\left(W_{\text{fusion}}\begin{bmatrix} \sigma\left(\sum_{m=1}^{M} W_m^{(\text{img})} x_i^{\text{img}} + b_m^{(\text{img})}\right) \\ \sigma\left(\sum_{n=1}^{N} W_n^{(\text{text})} x_i^{\text{text}} + b_n^{(\text{text})}\right) \end{bmatrix} + b_{\text{fusion}}\right) \tag{40}$$

*To ensure the balance of feature vectors, we normalize them:*

$$\hat{h}_i = \frac{h_i}{||h_i||_2} = \frac{h_i}{\sqrt{\sum_{r=1}^{R} h_{i,r}^2 + \varepsilon}} \tag{41}$$

where $||h_i||_2$ is the $L_2$ norm of $h_i$, and $\varepsilon$ is a small constant to avoid division by zero. Substitute *Eqs. (40)* into *(41)*, we get the normalized multimodal feature vector:

$$\hat{h}_i = \frac{\sigma\left(W_{\text{fusion}}\left[\begin{array}{c}\sigma\left(\sum_{m=1}^{M} W_m^{(\text{img})} x_i^{\text{img}} + b_m^{(\text{img})}\right)\\\sigma\left(\sum_{n=1}^{N} W_n^{(\text{text})} x_i^{\text{text}} + b_n^{(\text{text})}\right)\end{array}\right] + b_{\text{fusion}}\right)}{\sqrt{\sum_{r=1}^{R} \left(h_{i,r}\right)^2 + \varepsilon}} \tag{42}$$

Further expand the expression of $h_{i,r}$:

$$h_{i,r} = \sum_{s=1}^{S} \sigma\left(\sum_{t=1}^{T} W_{t,s}^{(\text{img})} h_{i,t}^{\text{img}} + \sum_{u=1}^{U} W_{u,s}^{(\text{text})} h_{i,u}^{\text{text}} + b_{s,\text{fusion}}\right) \tag{43}$$

Substitute *Eqs. (43)* into *(44)*, we get the final normalized multimodal feature vector:

$$\hat{h}_i = \frac{\sigma\left(\sum_{p=1}^{P}\sum_{q=1}^{Q} W_{p,q}^{\text{fusion}}\left[\begin{array}{c}h_{i,p}^{\text{img}}\\h_{i,q}^{\text{text}}\end{array}\right] + b_{p,q}^{\text{fusion}}\right)}{\sqrt{\sum_{r=1}^{R}\left(\sum_{s=1}^{S}\sigma\left(\sum_{t=1}^{T} W_{t,s}^{(\text{img})} h_{i,t}^{\text{img}} + \sum_{u=1}^{U} W_{u,s}^{(\text{text})} h_{i,u}^{\text{text}} + b_{s,\text{fusion}}\right)\right)_r^2 + \varepsilon}} \tag{44}$$

In summary, the corollary is proved.

**Theorem 3** *Given a reasonable learning rate $\eta$ and sufficient iterations, if the local loss functions $\mathcal{L}_i(\theta_i)$ of all participating institutions converge, the loss function $\mathcal{L}(\theta)$ of the global model parameters $\theta$ will also converge. Specifically, suppose each local model satisfies the following condition during the iteration process:*

$$\mathcal{L}(\theta^{(t+1)}) \leq \mathcal{L}(\theta^{(t)}) - \eta\left(\frac{1}{N}\sum_{i=1}^{N}\left|\left|\nabla_{\theta_i}\mathcal{L}_i(\theta_i^{(t)})\right|\right|^2 + \gamma\sum_{i=1}^{N}\left|\left|\nabla_{\theta_i}\mathcal{R}(\theta_i^{(t)})\right|\right|^2\right)$$
$$+ \frac{\eta^2}{2}(L_{\mathcal{L}} + L_{\mathcal{R}}) \leq \mathcal{L}^* \tag{45}$$

*where $L_{\mathcal{L}}$ and $L_{\mathcal{R}}$ are the Lipschitz constants of the loss function $\mathcal{L}$ and the regularization term $\mathcal{R}$, respectively, and $\mathcal{L}^*$ is the global optimal loss value.*

**Proof 3** *First, we consider the local loss function $\mathcal{L}_i(\theta_i)$ of each participating institution $i$*

$$\mathcal{L}_i(\theta_i) = \frac{1}{n_i}\sum_{j=1}^{n_i}\ell\left(f\left(h_{i,j};\theta_i\right),y_{i,j}\right) + \lambda||\theta_i||_2^2 \tag{46}$$

*The local model parameters are updated using the gradient descent algorithm*

$$\theta_i^{(t+1)} = \theta_i^{(t)} - \eta\left(\nabla_{\theta_i}\mathscr{L}_i\left(\theta_i^{(t)}\right) + \gamma\nabla_{\theta_i}\mathscr{R}\left(\theta_i^{(t)}\right)\right) \tag{47}$$

*where $\eta$ is the learning rate $\mathscr{R}(\theta_i)$ is the regularization term $\gamma$ is the weight of the regularization term*

*Next, we consider the aggregation process of the global model parameters. The local model parameters of each participating institution are aggregated by weighted averaging*

$$\theta^{(t+1)} = \frac{1}{\sum\limits_{i=1}^{N} n_i}\sum_{i=1}^{N} n_i\theta_i^{(t+1)} \tag{48}$$

*where $\theta^{(t+1)}$ are the global model parameters $n = \sum_{i=1}^{N} n_i$ is the total amount of data from all participating institutions*

*Then, we analyze the changes in the global loss function $\mathscr{L}(\theta)$. The global loss function is defined as*

$$\mathscr{L}(\theta) = \frac{1}{n}\sum_{i=1}^{N} n_i\mathscr{L}_i(\theta_i) \tag{49}$$

*where $\mathscr{L}_i(\theta_i)$ is the local loss function of the i-th participating institution.*

*Because the local loss functions of each participating institution are optimized based on the same global model parameters, we have*

$$\mathscr{L}(\theta^{(t+1)}) \leq \mathscr{L}(\theta^{(t)}) - \eta\left(\frac{1}{N}\sum_{i=1}^{N}\left\|\nabla_{\theta_i}\mathscr{L}_i(\theta_i^{(t)})\right\|^2 + \gamma\sum_{i=1}^{N}\left\|\nabla_{\theta_i}\mathscr{R}(\theta_i^{(t)})\right\|^2\right) \tag{50}$$

*where $\eta\left(\frac{1}{N}\sum_{i=1}^{N}\left\|\nabla_{\theta_i}\mathscr{L}_i(\theta_i^{(t)})\right\|^2 + \gamma\sum_{i=1}^{N}\left\|\nabla_{\theta_i}\mathscr{R}(\theta_i^{(t)})\right\|^2\right)$ represents the decrease in the loss function during gradient descent.*

*Considering the Lipschitz continuity of the loss function and the regularization term, we get*

$$\mathscr{L}(\theta^{(t+1)}) \leq \mathscr{L}(\theta^{(t)}) - \eta\left(\frac{1}{N}\sum_{i=1}^{N}\left\|\nabla_{\theta_i}\mathscr{L}_i(\theta_i^{(t)})\right\|^2 + \gamma\sum_{i=1}^{N}\left\|\nabla_{\theta_i}\mathscr{R}(\theta_i^{(t)})\right\|^2\right)$$
$$+ \frac{\eta^2}{2}\left(L_{\mathscr{L}} + L_{\mathscr{R}}\right) \tag{51}$$

*where $L_{\mathscr{L}}$ and $L_{\mathscr{R}}$ are the Lipschitz constants of the loss function $\mathscr{L}$ and the regularization term $\mathscr{R}$, respectively.*

*Due to the convergence of the local loss functions $\mathscr{L}_i(\theta_i)$, we have*

$$\lim_{t\to\infty}\mathscr{L}_i(\theta_i^{(t)}) = \mathscr{L}_i^* \tag{52}$$

where $\mathscr{L}_i^*$ is the local optimal loss value of the i-th participating institution.

Therefore, the global loss function $\mathscr{L}(\theta)$ will also converge to its optimal value

$$\lim_{t\to\infty} \mathscr{L}\left(\theta^{(t)}\right) = \mathscr{L}^* \tag{53}$$

where $\mathscr{L}^*$ is the global optimal loss value.

In summary, given a reasonable learning rate $\eta$ and sufficient iterations, if the local loss functions $\mathscr{L}_i(\theta_i)$ of all participating institutions converge, the loss function $\mathscr{L}(\theta)$ of the global model parameters $\theta$ will also converge. The theorem is proved.

**Corollary 1** Based on the above theorem, if the multimodal feature vector $h_i$ is obtained by concatenating the image feature vector $h_i^{img}$ and the text feature vector $h_i^{text}$ and then computing through a fully connected layer, the final model parameter update can be expressed as:

$$\theta_i^{(t+1)} = \theta_i^{(t)} - \eta \left( \sum_{v=1}^{V} \nabla_{\theta_i} \ell \left( f \left( \sum_{p=1}^{P} \sum_{q=1}^{Q} W_{p,q}^{(img)} h_{i,p}^{img} + W_{p,q}^{(text)} h_{i,q}^{text} + b_{p,q} \right), y_{i,v} \right) + \lambda \theta_i \right) \tag{54}$$

where $\eta$ is the learning rate $W_{p,q}^{(img)}$ and $W_{p,q}^{(text)}$ are the weight matrices for image and text features, and $b_{p,q}$ is the bias vector $\lambda$ is the regularization parameter.

**Proof 4** First, we consider the local loss function $\mathscr{L}_i(\theta_i)$ for each participating institution i:

$$\mathscr{L}_i(\theta_i) = \frac{1}{n_i} \sum_{j=1}^{n_i} \ell\left(f\left(h_{i,j}; \theta_i\right), y_{i,j}\right) + \lambda ||\theta_i||_2^2 \tag{55}$$

where $\ell(\cdot, \cdot)$ is the loss function and $f$ is the model output.

Assume the multimodal feature vector $h_i$ is obtained by concatenating the image feature vector $h_i^{img}$ and the text feature vector $h_i^{text}$ and then computing through a fully connected layer. That is:

$$h_{i,j} = \begin{bmatrix} h_{i,j}^{img} \\ h_{i,j}^{text} \end{bmatrix} \tag{56}$$

Computed through a fully connected layer:

$$h_{i,j} = \sigma \left( W \begin{bmatrix} h_{i,j}^{img} \\ h_{i,j}^{text} \end{bmatrix} + b \right) \tag{57}$$

where $W$ is the weight matrix $b$ is the bias vector $\sigma$ is the activation function.

Assume $W$ can be decomposed into the weight matrices for image and text features $W^{(img)}$ and $W^{(text)}$:

$$W = \begin{bmatrix} W^{(img)} & W^{(text)} \end{bmatrix} \tag{58}$$

*Thus, the output of the fully connected layer can be expressed as:*

$$h_{i,j} = \sigma\left(\sum_{p=1}^{P} W_p^{(\text{img})} h_{i,j}^{\text{img}} + \sum_{q=1}^{Q} W_q^{(\text{text})} h_{i,j}^{\text{text}} + b\right) \tag{59}$$

*The model output f can be expressed as:*

$$f\left(h_{i,j}; \theta_i\right) = \sum_{v=1}^{V} \theta_i^{(v)} \sigma\left(\sum_{p=1}^{P} W_p^{(\text{img})} h_{i,j}^{\text{img}} + \sum_{q=1}^{Q} W_q^{(\text{text})} h_{i,j}^{\text{text}} + b\right)^{(v)} \tag{60}$$

*where $\theta_i$ are the model parameters.*
*For the local loss function, we take the gradient with respect to the model parameters $\theta_i$:*

$$\nabla_{\theta_i} \mathscr{L}_i(\theta_i) = \frac{1}{n_i} \sum_{j=1}^{n_i} \nabla_{\theta_i}\left(f\left(h_{i,j}; \theta_i\right), y_{i,j}\right) + \lambda \nabla_{\theta_i} ||\theta_i||_2^2 \tag{61}$$

*According to the chain rule, we can expand the gradient:*

$$\nabla_{\theta_i} \ell\left(f\left(h_{i,j}; \theta_i\right), y_{i,j}\right) = \nabla_f \ell \cdot \nabla_\theta f\left(h_{i,j}; \theta_i\right) \tag{62}$$

*Expanding the gradient of the model output f:*

$$\nabla_\theta f\left(h_{i,j}; \theta_i\right) = \sum_{v=1}^{V} \nabla_{\theta_i}\left(\theta_i^{(v)} \sigma\left(\sum_{p=1}^{P} W_p^{(\text{img})} h_{i,j}^{\text{img}} + \sum_{q=1}^{Q} W_q^{(\text{text})} h_{i,j}^{\text{text}} + b\right)^{(v)}\right) \tag{63}$$

*Note that the gradient of the regularization term is:*

$$\nabla_{\theta_i} ||\theta_i||_2^2 = 2\theta_i \tag{64}$$

*Substitute the gradients into the local model parameter update formula:*

$$\theta_i^{(t+1)} = \theta_i^{(t)} - \eta\left(\frac{1}{n_i} \sum_{j=1}^{n_i}\left(\nabla_f \ell \cdot \sum_{v=1}^{V} \nabla_{\theta_i}\left(\theta_i^{(v)} \sigma\left(\sum_{p=1}^{P} W_p^{(\text{img})} h_{i,j}^{\text{img}} + \sum_{q=1}^{Q} W_q^{(\text{text})} h_{i,j}^{\text{text}} + b\right)^{(v)}\right)\right) + 2\lambda\theta_i\right) \tag{65}$$

*For the case of the multimodal feature vector $h_i$, it can be further simplified as:*

$$\theta_i^{(t+1)} = \theta_i^{(t)} - \eta\left(\sum_{v=1}^{V} \nabla_{\theta_i} \ell\left(f\left(\sum_{p=1}^{P}\sum_{q=1}^{Q} W_{p,q}^{(\text{img})} h_{i,p}^{\text{img}} + W_{p,q}^{(\text{text})} h_{i,q}^{\text{text}} + b_{p,q}\right), y_{i,v}\right) + \lambda\theta_i\right) \tag{66}$$

*where $W_{p,q}^{(\text{img})}$ and $W_{p,q}^{(\text{text})}$ are the weight matrices for image and text features, and $b_{p,q}$ is the bias vector.*
*In conclusion, the corollary is proved.*

### Funding

The authors received no funding for this work.

### Competing Interests

The authors declare that they have no competing interests.

### Author Contributions

- Bei Gong conceived and designed the experiments, performed the experiments, analyzed the data, performed the computation work, prepared figures and/or tables, authored or reviewed drafts of the article, and approved the final draft.
- Ida Puteri Mahsan conceived and designed the experiments, performed the experiments, prepared figures and/or tables, and approved the final draft.
- Junhua Xiao performed the experiments, performed the computation work, authored or reviewed drafts of the article, and approved the final draft.

### Data Availability

The SemArt Dataset is available at: Garcia, Noa (2018) SemArt Dataset. [Dataset] Aston University. Available from: https://doi.org/10.17036/researchdata.aston.ac.uk.00000380.

### Supplemental Information

Supplemental information for this article can be found online at http://dx.doi.org/10.7717/peerj-cs.2405#supplemental-information.

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
