# Peer review of "Federated learning-driven collaborative recommendation system for multi-modal art analysis and enhanced recommendations"

_PeerJ Computer Science, doi:10.7717/peerj-cs.2405_

## Round 0.1 · original submission · Major Revisions

Dear Authors,

Your paper has been revised. In view of the reviewers’ criticism, it needs major revision before being considered for publication in PeerJ Computer Science.

The principal issues that must be fixed are the following:

1) The structure and organization of the article have to be changed to present ideas more clearly. The article is challenging to follow. The motivation and context of the proposal should be stated clearly in the introduction. A related work section would give context to the proposed research.  A paragraph indicating the organization of the paper could be added at the end of the introduction. Also, a theoretical background section could include some detailed explanations of federated learning.

2) The paper must report comparisons among the authors’ algorithm and other related algorithms proposed in the literature.

3)The conclusions should be expanded to describe the limitations of the proposed approach, analyze implications for research and practice, and extend future research lines.

4) The raw data must be shared.

Reviewer 1 ·

Basic reporting

The article aims at presenting a framework for recommending art based on federated learning. The ideas presenting are interesting and the area of research is relevant. The proposal has been evaluated with several datasets.
However, the structure and organization of the article has to be changed in order to present ideas more clearly. The article is not easy to follow. The motivation and context of the proposal should be stated clearly in the introduction. A related work section would give context to the proposed research. Also, some detailed explanation of federated learning could be included in a theoretical bakcground section.

Experimental design

A paragraph describing the organization of section 3 should be included.
The research questions should be indicated clearly.
The authors should describe how experimental parameters were set.
The authors compare their algorithm with CNN and LSTM as baselines, but what about other related works in the literature?

Validity of the findings

The authors should analyze related works using federated learning and compare their approach against them, in order to assess the novelty of the proposal.
The conclusions should be enlarged describing the limitations of the proposed approach, analyzing implications for research and practice, and also extending future research lines.
The authors have provided the code.

Additional comments

- The introduction should state clearly the context and motivation of the proposed work.
- A paragraph indicating the organization of the paper could be added at the end of the introduction
- Table 1 should be moved to a related work section, that could also include detailed information about federated learning to reach a broader audience. Some ideas in section 2 could be moved to such related work section.
- Figure 1 should be described with more detail, depicting each part of the proposal
- Some paragraphs describing the contents of sections sold be included at the beginning of each of them
- Some theoretical background should be included before presenting the formulas in section 2.1
- The proposal is presented as a framework but then it is given in the form of an algorithm. The authors should clarify this issue.
- The conclusions should be enlarged with implications for research and practice.

·

Basic reporting

- Confirmed clear and unambiguous, professional English used throughout.
- Confirmed literature references, sufficient field background/context provided.
- Confirmed professional figures and tables.
- Some issues with article structure when compared to the PeerJ template. Some rearranging may be needed.
- Author uses "RESEARCH ISSUES" instead of "METHODS"
- Author uses "EXPERIMENTS AND RESULTS" instead of "RESULTS"
- Author uses "DISCUSSION" as a sub section
- Author uses "CONCLUSION" instead of "CONCLUSIONS"
- Confirmed raw data source shared and still available for download (3GB).
- Some issues with self-contained with relevant results to hypotheses. It's not clear how results map to a specific artist and artwork which is necessary for data privacy and copyright.
- Confirmed formal results should include clear definitions of all terms and theorems, and detailed proofs.

Experimental design

- Confirmed original primary research within Aims and Scope of the journal.
- Some issues with research question being well defined, relevant & meaningful.
- I am not sure that privacy applies as much as copyright. I think most artists want their work to be public and use an alias if they don't want their real name to be associated with the artwork.
- Challenges may exists for art institutions to produce/fund technical model generation and integration.
- Confirmed that it is stated how research fills an identified knowledge gap.
- Some issues with rigorous investigation performed to a high technical & ethical standard. Several things were not clear to me.
- The label of the accuracy measure is not clear. Is the label a style, artwork, artist name?
- The train, validation, and test percentages are not documented in the paper.
- Some issues with methods described with sufficient detail & information to replicate.
- train, validation, and test csv files missing. Do these define the train, validation, and test percentages?
- What hardware is used for experiment and how many images are actually used. The time to train the model seems lower than expected.
- Since text attributes will likely include artist name and artwork name, this will obviously increase accuracy.
- The experiment only uses data from SemArt which contains text features. It's likely that AI generated works will be image only which will omit any text related features.
- Only image folder observed in notebook. I am not sure how the text is loaded. Perhaps its in the same folder as images?

Validity of the findings

- Confirmed impact and novelty assessed.
- Due to time and some missing files, I am unable to confirm meaningful replication encouraged where rationale & benefit to literature is clearly stated.
- Confirmed that all underlying data have been provided; they are robust, statistically sound, & controlled.
- Some auxiliary is likely needed to reproduce.
- Some issues with conclusions are well stated, linked to original research question & limited to supporting results. Satisfying the questions above with related information documented in conclusion would satisfy this.

Additional comments

Thank you for sharing your work. I enjoyed reading your paper. I left some observations for your review. I apologize if some of my observations are in error.

Reviewer 3 ·

Basic reporting

At the start of the first paragraph of the introduction, there is a statement about how freelance artists and art galleries are the main creators and collectors of artworks. What about the private art collectors that usually purchase from galleries? Galleries usually purchase or exhibit artwork with the goal of selling it to a collector.

Text features are introduced in section 2.2.2, but very little detail is given on them. The text states that a transformer is used to extract the features, but more details on this would strengthen this portion.

The first paragraph of the discussion section (3.3) directly copies the introduction (1.1), word-for-word without one of the citations. I advise rewording one of the two sections.

Table 4 gives training time of the models in milliseconds. Often, a bit more detail is given about how that figure was reached.

Formatting issue on one of the references (Mintie 2023).

No raw data shared. There is a Jupyter Notebook given, and a link to the dataset is provided in the text of the paper, but the dataself itself is not shared.

Appears to be a margin issue with Table 1.

Experimental design

No comment.

Validity of the findings

No underlying data provided.

No other comments.

---

## Round 0.2 · Minor Revisions

Dear Authors,

Your paper has been revised. It just needs minor revisions before being accepted for publication in PEERJ Computer Science. Please address all the questions posed by reviewers before resubmitting the revised manuscript, highlighting all the changes you made to your work.

Reviewer 1 ·

Basic reporting

The article aims at presenting a framework for recommending art based on federated learning. The ideas presenting are interesting and the area of research is relevant. The proposal has been evaluated with several datasets.
The authors have addressed my concerns raised in the previous review, improving the article.

Experimental design

The authors have made changes to this section in a satisfactory way.

However, I suggest to indicate whether equations in the proposal are proposed by authors or if they are taken from other works, please add references to them.

Validity of the findings

Conclusions are rather short. I suggest to include limitations of the proposal and some lines describing future research lines.

Additional comments

My suggestions:
- Include at the end of the introduction a paragraph describing the organization of the article.
- Add references to equations taken from other words or indicate if they are proposed by the authors
- Enlarge the conclusions as indicated above.

Reviewer 3 ·

Basic reporting

No major comments. Clear and professional. Sufficient context.

Formatting issue on Mintie, K. "Mintie, K. (2023).
"A means of protection or destruction? ¡i¿copyright notifications on paintings in the399
united states, 1870-1911¡/i¿. AMERICAN ART, 37(3)." (Line 399 in the on page 14)

Experimental design

No comment.

Validity of the findings

No comment.

Additional comments

I would recommend mentioning the existence of wealthy art collectors and museums as well, who are also primary consumers of artworks. "Freelance artists and art galleries are the main creators and collectors of artworks."

---

## Round 0.3 · accepted · Accept

Dear Authors,
Your paper has been accepted for publication in PeerJ Computer Science. The comments of the reviewers who evaluated your manuscript are included in this letter. I ask that you make minor changes to your manuscript based on those comments, before uploading final files. Thank you for your fine contribution.

Reviewer 1 ·

Basic reporting

The authors have addressed my concerns and they have improved the article.

Experimental design

The authors have addressed my concerns in a satisfactory way.

Validity of the findings

The authors have addressed my concerns in a satisfactory way.

Additional comments

The article can be accepted for publication